# Preventing Overtourism by Identifying the Determinants of Tourists' Choice of Attractions

**Hugo Padrón-Ávila ***  **and Raúl Hernández-Martín**

Department of Applied Economics and Quantitative Methods, Universidad de La Laguna,
38200 San Cristóbal de La Laguna, Spain; rahernan@ull.edu.es
* Correspondence: hpadrona@ull.edu.es

**Abstract:** Popular tourism destinations based on specific attractions along with coastal and island destinations have been considered potential candidates to suffer from overtourism. In this context, in-depth knowledge of the determinants of tourists' choices of attractions can be used to improve policies against crowding. This paper analyzes why tourists decide to visit certain attractions instead of others in the context of an island destination with sustainability concerns. To do so, discrete choice models are used to determine if a set of 96 variables can explain why 11 attractions are visited on the island of Lanzarote. The results show that 86 variables are significant to explain visits to at least one of the attractions. The analysis also identifies both similarities and differences on the effects these variables have on the probability of visiting each of the 11 attractions. These results are useful to cluster attractions depending on the profile of those tourists most likely to visit them and to cluster variables regarding their effect on visiting attractions. Furthermore, the results provide useful information for public and private managers involved in evenly reallocating tourist flows in time and space to avoid the negative impacts of overtourism.

**Keywords:** attractions; overtourism; tourists' characteristics; destination management; discrete choice models

---

## 1. Introduction

Current media and academic debates have highlighted increasing concerns over the negative impact of the concentration of tourism flows in certain areas, particularly in cities, in a phenomenon that has been labeled as overtourism. Overtourism appears to be closely linked with the popularity of particular attractions and points of interest that have become 'hotspots' suffering from congestion [1]. However, effective attraction management can improve tourists' satisfaction and destination sustainability, avoiding the negative impacts of the growth of tourist flows. The effects of overtourism can also be often suffered on small tourism islands, where land availability is constrained [1].

The island of Lanzarote has an area of 846 km$^2$ and is the easternmost island of the Canary Islands, an autonomous region in Spain. The mild climate of these islands allows tourist arrivals throughout the whole year without a marked seasonality pattern. Lanzarote is the third main tourism island of the Canary Islands, which is the region with the highest figures of overnight stays in tourism accommodation in Europe [2]. Using data published by the Canary Islands Institute of Statistics for 2018, Lanzarote has a population of just 149,000 inhabitants and yet received 2.9 million tourists from outside the archipelago with an average length of stay of 7.8 days [3]. Lanzarote was declared a UNESCO Biosphere Reserve in 1993. Two years after this, the island hosted the World Conference on Sustainable Tourism [4], as part of its commitment to responsible tourism. However, the island has experienced tensions since the early economic growth of the 1960s between tourism development and the island's natural and cultural preservation [5]. In fact, the debate on the limits of growth and

sustainability began in the Canary Islands four decades ago, and especially in Lanzarote, this debate has been part of its strategy for tourism development for a long time. These concerns resulted in the implementation of a moratorium on new hotels in the Canary Islands in 2001 [6]. Despite these restrictions on the growth of the accommodation supply, tourism arrivals continued to increase in Lanzarote along with population figures (55% increase in inhabitants from 2000 to 2018) [3]. Therefore, public concerns with congestion and the negative impacts of rapid growth have been recurrent in the Canary Islands long before the recent renewed interest in tourism impacts under the concept of overtourism [7]. The interest of Lanzarote in sustainable tourism has been recognized widely and highlighted recently by research on sustainable tourism mobility on the island [8]. Furthermore, Eckert and Pechlaner recognized the conflict between sustainability and tourism growth in Lanzarote and made a proposal for product and target group market diversification [9]. These kinds of policies are closely related to the segmentation of tourists to improve destination management and, consequently, to the aims of this research.

The tourism model of Lanzarote is somewhat particular within the Canary Islands. Besides sun and sand facilities, the destination management organization is involved in providing a set of outstanding natural and cultural attractions to be visited by tourists. These attractions are mainly managed by a public-owned firm (Centros de Arte, Cultura y Turismo de Lanzarote—Arts, Culture, and Tourism Centers of Lanzarote), which depends on the Lanzarote Island Council. This set of attractions, including a volcanic national park, volcanic caves, cultural heritage attractions, etc., received 3 million visits in 2018, with a moderate 9.2% increase in visits to the main attractions between 2000 and 2018. Arts, Culture, and Tourism Centers of Lanzarote obtained a €30 million turnover in 2018, of which 67.3% came from selling tickets, 20.7% from restaurant sales, and 11.1% from sales made in the shops of the attractions [10].

As mentioned, concerns about congestion arose in Lanzarote at least four decades ago, long before the recently renewed academic interest in the negative effects of rapid tourism growth, fostered by low-cost airlines, the sharing economy, new booking platforms, social networks, etc., put the topic in the media and the academic spotlight [11]. However, the island of Lanzarote can be an example of successful public management of a destination and its attractions. In fact, a survey on 'Island Affairs' conducted in December 2017 for the UNESCO Biosphere Reserve shows a remarkable improvement from 2006 to 2017 in inhabitants' perceptions about the environmental situation of the island [12].

Researchers have tried to understand tourists' mobility patterns for a long time [13], particularly those tourist tracks connecting accommodation and the main attractions of the destinations. Despite this being a tough task, Shoval and Isaacson [14] point out that simply identifying a destination's attractions and locating them on a map provides an analysis of how tourists may move between these attractions. These kinds of studies can have great relevance for both destination managers and tourism attraction firms. In fact, Liu, Huang, and Fu [15] stated that tourism movements and flows are some of the most important factors in creating connections between tourism attractions. In spite of the relevance of these studies for destination competitiveness and sustainability, Shoval et al. [16] pointed out that there has been little research on this subject. The problem of these analyses lies in the complexity of these movements and the relationships they generate [17]. This difficulty derives not only from the variety of itineraries and places to visit that tourists can choose from [18], but is also influenced by tourists and trip characteristics [19] and by the location of the key elements that make up the destination, such as facilities, attractions, tourist accommodation, etc. [20].

In this paper, the aim is to determine which tourist characteristics make them more likely to visit certain attractions instead of others. This information is crucial for the management of attractions to prevent overtourism. To do so, several binomial discrete choice logit models have been used. These provide an understanding of how a set of 96 different variables impacts the probabilities of visiting 11 different attractions.

First, a literature review was carried out to discover which variables were pointed out by other researchers as determinants to explain tourists' visits to attractions, methods used to analyze these

data, and the relevance of the concept of overtourism in the context of a mass tourism destination based on attractions. After this, the data and the method chosen to carry out the study are explained. Afterwards, the results of the research are presented in a table that contains data regarding the extent that the variables used change the probabilities of visiting each of the 11 attractions studied. The data shown in the table are discussed to establish which variables explain why each of the attractions is visited or not. Finally, the conclusions of the research are presented and the results obtained are compared with previous research. Knowing which variables explain visits to attractions is useful for destination managers aiming to prevent overtourism and to design revitalization policies.

## 2. Literature Review

Overtourism is not a recent phenomenon [7,11,21]. Nevertheless, the debate on overtourism has only attracted the interests of academics in recent years. Although this debate has often been located in cities and urban tourism, Koens et al. [11], Oklevik et al. [22], or Capocchi et al. [23] clarified that overtourism impacts are not necessarily limited to urban settings. Thus, while reports on overtourism by UNWTO [24,25] focused on urban contexts, Peeters et al. applied the concept to four types of destinations: urban; coastal and islands; rural; and heritage and attractions [1]. The relevance for island destinations was also stated by Koens et al. [11]. Indeed, overtourism can be relevant to analyze the dynamics and management of places such as Lanzarote, an island destination with problems of congestion where the tourist experience is based on visits to specific attractions related to nature and heritage. Nevertheless, in their exhaustive revision of the literature, Capocchi et al. [23] show that the concept of overtourism is neither a new phenomenon nor one that is well defined.

Koens et al. [11] showed that the issues related to overtourism had already been discussed in earlier studies. They provided seven aspects related to overtourism to allow a better understanding of this phenomenon. Among the issues they highlighted, there was one with particular relevance for this research: its impacts are not city-wide because they can be concentrated in any area with special tourist attractiveness. As analyzed by Oklevik et al. for the case of Fjord Norway [22], the problems associated with increasing tourist flows are related to the tourist activities at certain points of interest. This is why authors such as Neuts and Nijkamp [26] or the UNWTO [25] call for policies aimed at spreading tourist flows in time and space. This spreading of flows has as a starting point a segmentation of tourists based on their characteristics, as proposed in the current research.

Recent studies have identified certain factors that explain the places visited by tourists according to tourists' characteristics. Lew, Hall, and Williams [27] suggested that the most relevant variables to explain visitors' behavior are motivational factors. Lew, Hall, and Williams [27] also pointed out that these factors affect the choice of the destination and the attractions visited. This hypothesis was supported by carrying out a bibliographic review and an empirical study that showed that the elements that explain the attractiveness of an attraction depend on the thoughts and expectations tourists have before visiting the attraction. In fact, McKercher and Koh [28] noted that many tourists are not attracted to a tourist attraction in itself, but rather because some attractions can help them eliminate stress, spend quality time with their families, keep up with friends, or to have an adventure. Winter [29] also analyzed the influence of motivations and concluded, for example, that tourists who stated an interest in culture were more likely to enjoy the cultural resources of a destination. In addition, Dai, Hein, and Zhang [30] also pointed to tourists being interested in local culture as a key element to explain tourists' behavior, as these type of tourists have particular and different interests.

However, some studies have shown that there are other variables that directly affect tourists' visiting patterns. In particular, it seems that a trip's characteristics affect the probabilities of visiting certain places instead of others. Müller [31] observed that those visitors who travel to destinations where they have second homes do not show much interest in traveling widely throughout the destination, but visit places near their accommodation. Getz [32] analyzed, for example, the places MICE tourists tend to visit in comparison to the places visited by other kinds of visitors. The conclusion of the study was that MICE tourists are more likely to visit exclusively places linked to the purpose of their trip, while

other segments visit a wider set of attractions. Similarly, Ormond [33] pointed out that visitors who travel for health tourism purposes do not tend to visit places that are not linked to the main purpose of their trip, so this segment bases its trips on going to hospitals, health and welfare centers, clinics, etc.

Other studies have shown that visitors' motivations and trip characteristics are not the only factors that condition their visits to attractions. Certain studies have highlighted the relevance of the socio-economic characteristics of visitors and attractions when it comes to explaining the type of places visited by tourists [34–36]. Timothy [37] stated that price is one of the factors that influences visitors' movements. His study analyzed the link between luxury destinations and their attractiveness for the segment of tourists with the greatest purchasing power. Other socioeconomic variables, such as gender or age, were also proven to have an influence on tourists' choices [38]. Visser [39] and Waitt and Markwell [40] also studied the effect of sexual orientation as one of the factors that influences the interests and motivations of visitors and its influence on visits to certain places during their trips. Bo et al. [41] and Park et al. [42]analyzed the effect of tourists' country of origin to study how this factor influences movements. The results of these studies show that foreign visitors seem to have a greater tendency to visit the most well-known tourist attractions of a destination, whereas national and regional visitors are more likely to move more actively within the destination and to spend larger amounts of money at more distant and less known attractions. In fact, Molinillo and Japutra [43] analyzed the variables conditioning domestic tourists' visits to cultural attractions in Andalusia, as these tourists have different reasons to choose the attractions and destinations to visit [43,44]. Moreover, Shoval et al. [16] stated that the location of the accommodation also conditions the places visited, since tourists have greater probabilities of visiting attractions located near their accommodation establishment. In addition, the characteristics of attractions and visits themselves are also key factors that could be analyzed to explain why some tourists are more interested in certain attractions than in others [45,46]. As a conclusion, it seems that a large number of variables have an influence on the willingness of tourists to visit attractions and recent studies are just starting to figure out some of these variables, but future research will probably enhance this list of determinants. In Table 1, the main findings of the literature review carried out are summarized.

**Table 1.** List of variables explaining tourists' choices.

| Variable | Reference |
|---|---|
| Socio-economic characteristics | [34–36] |
| Age | [38] |
| Gender | [38] |
| Country of residence | [41–44] |
| Sexual orientation | [39,40] |
| Motivations | [27,28] |
| Relaxation | [28] |
| Culture | [29,30] |
| MICE tourism | [32] |
| Health | [33] |
| Characteristics of attractions | [45,46] |
| Price | [37] |
| Second homes | [31] |
| Accommodation location | [16] |

There are several techniques that can be used to determine which variables explain the choice of attractions by tourists. In fact, the research carried out by Wang et al. [47] established numerous measures that can be used to identify the variables that encourage tourists to go to a specific place during their trip. In addition, it seems to be possible to use both qualitative and quantitative variables for this type of analysis [48,49]. However, despite qualitative data being used to carry out wider research, the process of compiling and analyzing this kind of data is more complicated. The reason is that the data gathered by qualitative techniques may differ from one visitor to another, making it

difficult to compare and obtain conclusive results. Having quantitative variables, on the other hand, makes it possible to carry out econometric models, which Kanji [50] stated to be a more suitable way to measure phenomena.

Among others, cluster analysis, structural equation modeling, linear regression, and discrete choice models were used to explain which variables impact tourists' behavior. Cluster analysis is a suitable technique for grouping tourists' characteristics in order to segment tourism demand, as it is possible to analyze which activities are more likely to be done by each segment, as shown by Scuderi and Dalle Nogare [51]. However, this technique does not identify the variables explaining the behavior studied or specify the relevance of each variable. By contrast, the use of structural equation modeling can be used to find out which variables explain why tourists behave the way they do. Despite this, it can only be applied in studies using primary Likert scale data to estimate the results [52,53], and these models have problems to show results when there are omitted data or where dichotomic variables are used [54]. Regarding linear regression, studies like the one carried out by Marrocu, Paci, and Zara [34] used this technique to understand tourists' behavior, consumption, or intentions. Although linear regression can determine the variables explaining behavior, it can only be used to analyze linear relationships. Moreover, it requires a mean error of zero, a constant variance of the error, and that there is no autocorrelation between errors [55,56]. In the case of discrete choice models, they cannot indicate to what extent the variables analyzed determine the behavior studied when data contain non-binary variables [57–60]. Nevertheless, several studies, such as the ones carried out by Masiero and Zoltan [61]; Li, Yang, Shen, and Wu [62]; and Asero and Tomaselli [63], used this technique to study the variables explaining why tourists chose to visit certain places. However, they do not identify to what extent the behavior is explained by the variables analyzed. To do so, average marginal effects must also be calculated [59,64,65].

## 3. Method

The purpose of this research is to identify which tourist characteristics condition visits to different attractions in order to provide useful information for destination management and the prevention of problems linked to overtourism. The data to carry out this research were obtained from the responses provided by tourists in the Tourism Expenditure Survey conducted by the Canary Islands Institute of Statistics (ISTAC). This survey was carried out at Canary Island airports on tourists' returning home. The survey conducted on the island of Lanzarote contains a question regarding visits to 11 tourist attractions on the island that were selected by the destination management organization. The list of attractions includes: Jameos del Agua (volcanic cave and art museum), Timanfaya National Park (volcanic area), Los Verdes Cave (volcanic cave), Gazer del Río, Monument to the Peasant (history museum), San José Castle (modern art museum and castle), Cactus Garden, César Manrique Foundation (museum), Arrecife Town, the markets of Teguise and Haría, and La Graciosa (a small island close to Lanzarote) [66]. Of these 11 points of interest, the first seven were attractions managed by the Arts, Culture, and Tourism Centers of Lanzarote, while the rest included a private foundation displaying the legacy of César Manrique (the renowned artist and environmental activist of the island), who supported a sustainable tourism model for the island [67]. The other places of interest were the island's capital, the public markets of two municipalities, and the small island of La Graciosa, which is 7 km from the northern village of Órzola and accessible by ferry. Of the seven attractions analyzed and managed by Arts, Culture, and Tourism Centers of Lanzarote, six charge entrance fees from €4 to €10.

In previous research by Padrón-Ávila and Hernández-Martín [68], a study was carried out to understand which variables made tourists more likely to visit at least one of the attractions included in the list of attractions in the survey. However, in this study the analysis was conducted to determine which variables determine the visit to each of the 11 attractions analyzed. This analysis is relevant for the management of tourism flows to the main points of interest of the island and to optimize tourism from an economic, social, and environmental point of view, as proposed by Oklevik et al. [22].

First, a pilot study using contingency tables was applied to know which variables could explain whether the attractions analyzed were visited or not. Afterwards, 96 variables were chosen to carry out the analysis based on the results obtained and the literature review presented in the previous section. The choice of the variables was based on the research by Padrón-Ávila and Hernández-Martín [68]. The variables chosen were: tourists' gender, age, annual incomes, country of residence, main reasons for travelling, motives for choosing Lanzarote as a destination, overnight stays, travel group, purchase of low-cost flights, purchase of all-inclusive packages, repetition at the destination, total expenditure at origin, total expenditure at the destination, practice of diving, and overall assessment of the trip. Moreover, the 11 variables regarding visits to tourist attractions in Lanzarote were used to determine if the visit to a given attraction is linked with a visit to others. Most variables used in the analysis were binary variables (take the values 0 or 1). In the case of tourists' country of residence, 20 binary variables attached to different countries were used. In the case of main reasons for travelling, six binary variables were used to indicate if tourists were travelling for holidays, business, attending conferences, health, visiting friends and relatives, or other motives. In the case of motives to choose Lanzarote as a destination, 23 different motives were considered and tourists could indicate up to three in the survey. In the case of travel group, six different groups were considered, as tourists could indicate if they were travelling with their partner, with children, with other relatives, with friends, or with other companions. Variables regarding tourists' age, overnight stays, and expenditure were continuous. In the case of expenditure at the destination, this variable was divided into 19 items regarding different services and products typically bought by tourists. Moreover, tourists were divided into seven discrete ranges according to their annual incomes, so a lower value in this variable implies a lower income perceived by the respondent. In addition, a Likert scale of five items was used to measure the variable regarding the assessment of the trip made.

The complete database given by ISTAC contains 257,687 survey responses gathered from tourists visiting any of the Canary Islands. As this research focuses on 11 attractions located on the island of Lanzarote, a total of 43,301 responses of tourists that were surveyed from January 2010 to December 2016 at the airport of Lanzarote were considered. However, 14.42% of tourists surveyed in Lanzarote did not respond to the question regarding the attractions visited or not visited on the island. Therefore, these responses were not considered when analyzing an attraction for which the corresponding data had not been provided. Regarding the attractions analyzed, four of them were visited by more than a quarter of tourists surveyed. There were 48% of tourists who visited the Timanfaya National Park, 37% visited Jameos del Agua, 31% went to Gazer del Río, and 29% went to Los Verdes Cave. Five attractions were visited by 10%–25% of tourists: these attractions were the Cactus Garden, Arrecife Town, the traditional markets (these ones were visited by 21% of tourists), the Monument to the Peasant (14%), and the César Manrique Foundation (13%). San José Castle and La Graciosa were visited by just 5% of tourists surveyed.

The Survey on Tourism Expenditure of the Canary Islands included 43,301 responses to the survey in the island of Lanzarote from January 2010 to December 2016. The survey was conducted in the departure hall of the airport following international recommendations for tourism expenditure statistics [69], and it used non-probability stratified sampling in three stages. This method does not allow for obtaining sample errors. In the first stage of the sampling process, the week with the most air traffic was chosen in each month of the year. In a second stage, stratified sampling allowed to choose the countries and airports of origin of tourists and the timetables. In the third stage, tourists from mainland Spain and foreign countries were selected [70].

Literature on sampling states that biases and sample variance are the two possible errors when using non-probability sampling. To prevent these errors from affecting the results, the methodology applied in the Survey on Tourism Expenditure of the Canary Islands included elements of control and correction. To avoid biases, a post-stratification weighting was conducted with the help of auxiliary information to correct biases in age, gender, country of residence, and type of accommodation; thus reliability of estimates and inferences was increased [71]. To control for sampling variance, a minimum

of 20 observations was required for each variable. This threshold settled a maximum admissible error under assumptions of probabilistic sampling and maximum population variance. In this study with a sample of 43,301, this threshold was exceeded by far in the responses.

Binomial discrete choice logit models and average marginal effects were used to analyze data. Discrete choice models identify which variables are significant to explain why a certain attraction is visited [57,59,60], while average marginal effects show to what extent a given characteristic increases or decreases the probabilities of visiting an attraction [59,64,65]. Once the logit models for each attraction had been carried out, the analyses were repeated omitting non-significant variables and variables not analyzed due to contingency errors to check the results obtained. When repeating the analysis for the first time, all variables with a Z statistic over 1.50 were used, even though a minimum 1.96 Z statistic is needed to be considered significant. The reason to do so was that the results of these models depend on the independent variables selected by the researcher [57–60]. Therefore, not using independent variables with a Z statistic under 1.50 could increase the significance of other variables. Thus, a second model was carried out for each attraction with those independent variables that had a Z statistic over 1.50 in the results of the first model. In this second analysis, some variables did not show to be significant in the case of all the attractions. For this reason, non-significant variables were removed, and a third analysis was carried out. In the case of some attractions, this third model also showed non-significant variables; hence, they were removed, and the model was repeated until it was possible to carry out a model where all variables were significant. The analyses of San José Castle and Cactus Garden were carried out four times to achieve a model where all independent variables were significant, whereas the analyses of César Manrique Foundation and the town of Arrecife were performed five times. The others were run just three times.

Once the discrete choice models were calculated and the significant variables to explain tourists' behavior had been identified, average marginal effects of these variables were calculated for each attraction. Calculating the average marginal effects of these models, the results showed the increase or decrease in the probability a tourist has of visiting an attraction when the tourist shows a value of 1 for one of the binary variables analyzed [59,64,65]. In addition, when analyzing other kinds of variables (continuous, Likert scale, or divided by ranges), the results indicated how the average probabilities of visiting an attraction were modified by each unit added to the variables studied. For example, in the case of age, results of average marginal effects showed to what extent probabilities of visiting an attraction are modified by each extra year of a tourist's age, taking 16 years of age as the starting point, as this was the age of youngest tourists surveyed. If the coefficient resulting from average marginal effects was positive, it meant that each extra unit added to the variable increased the probabilities of visiting the attraction analyzed, while the probabilities would be reduced by each extra unit added to the variable if the result was negative. To carry out both analyses (discrete choice models and average marginal effects), the software Stata version 12 was used. As the results of average marginal effects are coefficients, these coefficients were converted into percentages in this paper to facilitate the interpretation of the results.

## 4. Results

In Table 2, it is possible to observe the results obtained for the average marginal effects of the discrete choice models used. Of the 96 original independent variables, nine were not significant for explaining whether any of the attractions studied were visited or not. For this reason, these variables were omitted and do not appear in Table 2. The variables deleted were: resident in Czech Republic, resident in Luxembourg, resident in another country not specified in the survey, having a motivation to travel not specified in the survey, choosing the destination for reasons not specified in the survey, leaving the choice of destination to another person, travelling with workmates, spending on personal services, and spending on investments. The other 86 independent variables were significant to explain at least why one of the eleven attractions was visited. In addition, four variables had a positive effect on the visit to all the attractions studied. These variables were: visiting the Cactus Garden, César

Manrique Foundation, Arrecife, and the markets of Teguise and Haría. They are all variables used to find out if visiting an attraction increases the probabilities of visiting another attraction.

**Table 2.** Average marginal effects of binomial discrete choice logit models applied to the determinant variables that explain the visit to 11 tourism attractions in the island of Lanzarote.

| Variables | Jameos del Agua | Timanfaya | Los Verdes Cave | Gazer del Río | Monument to the Peasant | San José Castle | Cactus Garden | César Manrique Foundation | Arrecife | Teguise and Haría Markets | La Graciosa |
|---|---|---|---|---|---|---|---|---|---|---|---|
| Gender | | | | | 1.00 | | | −1.44 | 1.49 | | |
| Age | −0.10 | −0.11 | −0.04 | | 0.04 | | 0.08 | | 0.26 | 0.13 | −0.04 |
| Spain | 8.57 | | 10.28 | | 6.80 | −0.78 | −5.74 | | | | 4.86 |
| Germany | | −4.77 | 1.49 | | 4.79 | −2.53 | | 11.30 | | 5.74 | |
| Austria | | −8.30 | | 8.15 | 6.63 | −2.28 | | 8.32 | | | |
| Belgium | | | | −7.43 | 7.42 | | −4.24 | 4.56 | −5.49 | | |
| Denmark | | −7.64 | | | | | | | −9.36 | | |
| Finland | | −12.03 | | | | | −11.59 | | 5.56 | | |
| France | | | 2.51 | | | | 4.94 | | −4.81 | | 5.11 |
| Netherlands | | | 2.24 | −17.02 | 13.66 | | −5.10 | | | | −3.28 |
| Ireland | −4.50 | −9.78 | | −7.88 | | | −7.29 | | −6.50 | | −2.78 |
| Italy | | | 2.37 | | 5.88 | −1.87 | | | | 6.22 | 5.62 |
| Norway | −5.77 | −9.92 | | | | | −4.62 | | | | |
| Poland | | | | | | | | 11.84 | | | |
| Portugal | | | | −7.77 | 8.94 | | | | | | 9.24 |
| United Kingdom | −3.87 | −6.89 | | −5.95 | | | −6.60 | | −9.09 | | −2.62 |
| Russia | | | 14.37 | | | | −6.04 | | | | |
| Sweden | −4.59 | −9.32 | | | | | | 4.66 | | | |
| Switzerland | | | | | | | | 6.33 | | | |
| Incomes | −0.25 | 0.51 | | 0.45 | | | | 0.48 | | −0.85 | |
| Jameos del Agua | | 13.47 | 15.84 | 12.35 | 4.77 | | 9.44 | 8.33 | 3.34 | | 1.51 |
| Timanfaya | 11.69 | | 12.60 | 9.76 | 5.74 | | 6.26 | 5.29 | 5.86 | 6.15 | 1.48 |
| Los Verdes Cave | 12.74 | 11.96 | | 10.24 | | 1.95 | 7.29 | 2.14 | 2.64 | 3.25 | 1.08 |
| Gazer del Río | 10.60 | 9.95 | | | 4.28 | 2.83 | 2.43 | 3.91 | 6.24 | 6.29 | |
| Monument to Peasant | 5.01 | 7.86 | | 5.48 | | 2.98 | 3.45 | 6.74 | 4.60 | 3.35 | 1.54 |
| San José Castle | | | 6.76 | 7.95 | 6.53 | | 10.60 | 6.19 | 18.74 | 3.86 | 2.07 |
| Cactus Garden | 8.82 | 6.47 | 8.07 | 1.65 | 3.03 | 3.87 | | 7.86 | 3.59 | 2.55 | 1.09 |
| César Manrique Foundation | 7.47 | 3.54 | 3.51 | 4.77 | 5.35 | 2.23 | 7.51 | | 2.89 | 8.24 | 2.23 |
| Arrecife | 1.74 | 3.56 | 2.66 | 4.61 | 2.70 | 4.18 | 2.94 | 2.59 | | 15.53 | 1.47 |
| Teguise and Haría Markets | 1.35 | 3.84 | 3.62 | 4.52 | 1.97 | 0.97 | 2.11 | 4.90 | 14.97 | | 2.74 |
| La Graciosa | 1.81 | | 2.10 | | 2.42 | 0.91 | 1.78 | 3.56 | 3.75 | 7.04 | |
| Diving | | | 2.93 | | 2.60 | 1.46 | | | 5.35 | 5.81 | 5.57 |
| Motivation: Holidays | | | | | −6.69 | −2.16 | | | | | |
| Motivation: Business | | −10.60 | | −10.17 | | | | | | | −4.99 |
| Motivation: MICE | | | | −14.46 | | | | | | | |
| Motivation: Health | | | | | | | 8.69 | | | | |
| Motivation: Family | | −9.37 | | −8.08 | | | | | | | |
| Motive: Weather | | | | | | | | | | | −1.23 |
| Motive: Beaches | | | | | | | −1.26 | | | 1.78 | 1.67 |
| Motive: Landscapes | 1.77 | 5.40 | 2.37 | 3.07 | 1.28 | 0.64 | 1.22 | | | 2.04 | |
| Motive: Environment | | | | 2.04 | | | | | | | |
| Motive: Relaxation | 0.87 | | | | | | | | −1.09 | 1.57 | |
| Motive: Safety | | | | −1.44 | | | | | | | |
| Motive: Culture | 4.00 | 4.93 | | | | 1.42 | 2.23 | 5.42 | | 2.85 | |
| Motive: Rural tourism | | | | 4.13 | | | 2.98 | | | 6.19 | |
| Motive: Active tourism | | 3.20 | | 2.36 | | | | | | | 4.56 |
| Motive: Health tourism | | | | | | | | | 7.14 | | |
| Motive: Theme parks | | | | 7.70 | | | | | | | |
| Motive: Sea activities | 3.29 | | | | −2.82% | | | | −4.05 | | 2.03 |
| Motive: Golf | | | | | | | | | | | −10.66 |
| Motive: Other sports | −3.88 | | | 3.99 | | | | | −4.65 | | |
| Motive: Nightlife | | | | −5.34 | −4.91 | | | −4.05 | −4.04 | −3.98 | |
| Motive: Purchasing | | | | −4.32 | | | | −7.58 | | 5.42 | |
| Motive: Visit new place | 2.06 | 4.35 | 2.43 | 1.35 | | | | −1.23 | | 3.15 | |
| Motive: Easy transport | | | | | | | | | | | −1.44 |
| Motive: Price | | 2.41 | | | | | | | | 2.61 | |
| Motive: Suitable children | | | | | | | | | −3.37 | | −2.15 |
| Motive: Natural safety | −8.66 | | | | | | | | | | |

**Table 2.** *Cont.*

| Variables | Jameos del Agua | Timanfaya | Los Verdes Cave | Gazer del Río | Monument to the Peasant | San José Castle | Cactus Garden | César Manrique Foundation | Arrecife | Teguise and Haría Markets | La Graciosa |
|---|---|---|---|---|---|---|---|---|---|---|---|
| Group: Couple | | 2.01 | | | | | | | | | −1.44 |
| Group: Children | | 3.93 | | −1.46 | −1.66 | | −0.86 | | −3.59 | | −1.90 |
| Group: Relatives | | | | | | | | 2.71 | | 2.26 | −1.84 |
| Group: Friends | | 3.14 | | 1.86 | | | | | | | |
| Group: Others | | −21.71 | | | | | | | | | |
| Repetition | −1.11 | −8.01 | −1.36 | | | | | | | | 1.13 |
| Overnights | | −0.15 | | | | | | | 0.37 | 0.28 | 0.10 |
| All-inclusive | | −3.39 | | −2.52 | −1.35 | | | −3.04 | −4.20 | | −1.38 |
| Low-cost flight | | | | | | | 0.89 | | 3.50 | 1.30 | |
| Expenditure: Origin | | | | | | −0.00 | | | −0.00 | | |
| Expenditure: Destination | 0.00 | 0.00 | 0.00 | | | | | 0.00 | 0.00 | 0.00 | 0.00 |
| Expenditure: Hotel | −0.03 | | | | | | | | | | −0.06 |
| Expenditure: Extras | | | −0.06 | | | | | | | | −0.09 |
| Expenditure: Bus | −0.17 | | | | | | −0.07 | | 0.45 | 0.23 | |
| Expenditure: Taxi | −0.11 | −0.19 | −0.15 | −0.21 | | | −0.12 | | | −0.15 | −0.07 |
| Expenditure: Car renting | | 0.17 | 0.08 | 0.17 | | | 0.07 | | 0.10 | 0.04 | −0.05 |
| Expenditure: Excursions | 0.11 | 0.38 | | | | | | 0.08 | −0.10 | | |
| Expenditure: Leisure | | 0.13 | 0.12 | | | | | | 0.08 | | −0.09 |
| Expenditure: Other islands | | | | | | | | | | | 0.35 |
| Expenditure: Sport | −0.08 | | | | | | | | | | −0.12 |
| Expenditure: Culture | 0.13 | 0.34 | 0.19 | | | | | 0.26 | | | −0.08 |
| Expenditure: Health | | | −0.10 | | | | | | 0.10 | | |
| Expenditure: Supermarket | −0.06 | −0.05 | −0.02 | | | | | −0.02 | | | −0.05 |
| Expenditure: Discos | −0.10 | −0.11 | −0.08 | | | | | | | | −0.09 |
| Expenditure: Souvenirs | −0.04 | | | | | | | | | 0.07 | −0.08 |
| Expenditure: Restaurants | −0.04 | | | | | | | | | | −0.06 |
| Expenditure: Pharmacies | −0.17 | | | | | | | | | | −0.18 |
| Expenditure: Others | −0.07 | | | | | | | | 0.05 | | −0.08 |
| Trip assessment | | 1.74 | | | | −0.42 | | 0.76 | −0.94 | | 0.89 |

The results obtained show that the attractions studied were affected in the same way by several variables. This means that when a variable was significant to explain why two or more attractions are visited, the effect of that variable will be positive for all the attractions or negative for all of them. For example, tourists living in Denmark, Ireland, Norway, or the United Kingdom are less likely to visit the attractions studied in the survey. The same happened with tourists who travel for a holiday, who travel for business, and those who travel to visit relatives. However, tourists choosing the destination for its nightlife or because it is suitable for children showed a high propensity to visit the attractions contained in the survey. Purchasing all-inclusive packages for travelling to Lanzarote reduced the probabilities of visiting almost all the attractions included in the survey. In fact, Lew and McKercher [18] already pointed out that tourists purchasing all-inclusive packages are less likely to visit attractions. Moreover, each extra euro spent on booking the trip at the origin, at the accommodation establishment, buying extras in hotels, paying taxis, sport activities, supermarkets, night clubs, restaurants, and pharmacies also decreased the odds of visiting attractions on the island. By contrast, visiting any of the 11 attractions contained in the survey increased the probabilities of visiting the other attractions analyzed. In addition, choosing the destination for its landscapes, environment, culture, rural tourism products, active tourism activities, or for its price also increased the probabilities of visiting these attractions. Travelling with friends, travelling with low-cost flight companies, and each extra euro spent at the destination also had a positive effect on the increase of the probabilities of visiting attractions in Lanzarote.

Despite most variables showing an effect on explaining the visit to two or more attractions, 13 of them just explained why one of the attractions studied was visited. Of these 13 variables, six had a positive effect on the visit to the attraction and seven of them reduced the probabilities of visiting the attraction. Tourists coming from Poland had an 11.84% higher probability of visiting the Cactus

Garden. Moreover, tourists who travelled for health purposes were also 8.69% more likely to visit this attraction. Tourists coming from Switzerland showed a 6.33% higher tendency of visiting the César Manrique Foundation. Choosing the destination for health tourism activities or due to its theme parks increased the odds of visiting Arrecife and Gazer del Río, respectively, by just over 7%. Each extra euro spent visiting other islands of the archipelago also increased the probabilities of visiting the island of La Graciosa by 0.35%. However, traveling to attend congresses or meetings reduced the odds of visiting Gazer del Río by 14.46%. In addition, choosing Lanzarote as a destination for its weather or due to the ease of going to Lanzarote reduced the probabilities of visiting La Graciosa by just over 1%. Choosing the destination for the possibility of playing golf also reduced the probabilities of visiting this attraction by 10.66%. If the motive for choosing Lanzarote as destination was its security, the probabilities of visiting the Monument to the Peasant were reduced by 1.44%. But if the motive was its safety against natural disasters, the odds of visiting Jameos del Agua decreased by 8.66%. Finally, if the tourist travelled with types of companions not included in the survey, the probabilities the tourist had of visiting Timanfaya National Park were reduced by 21.71%. In fact, previous studies already indicated that the company during a trip is key to determine which attractions will be visited [72].

The other 33 variables pointed out as significant by the discrete choice models influenced the probabilities of visiting at least two of the eleven attractions studied. However, they had a positive effect on some attractions and a negative one on others. Likewise, some of these variables did not show a very significant effect on the modification of the probabilities, while others had a higher relevance. For example, gender seems to be a variable with only a slight effect on visits to attractions in Lanzarote. Male tourists were 1% more likely to visit the Monument to the Peasant and 1.49% more likely to visit Arrecife than female ones, whereas female tourists only had a 1.44% higher probability of visiting the César Manrique Foundation. Age showed a more significant effect. Each extra year of a tourist's age, made that tourist less likely to visit attractions linked with nature and sport, such as Jameos del Agua, Timanfaya, Los Verdes Cave, or La Graciosa. However, older tourists had greater odds of visiting cultural attractions such as the Monument to the Peasant, Cactus Garden, César Manrique Foundation, and the traditional markets of Teguise and Haría. Another example is tourists' annual incomes, which also seem to be a relevant variable to explain why certain attractions were visited. On the one hand, tourists with higher salaries were over 0.4% more likely to visit Gazer del Río and the César Manrique Foundation per each higher income range used in the survey (there were six possible ranges; more information in the Methodology Section). On the other hand, tourists with lower incomes were more likely to visit Jameos del Agua and the markets of Teguise and Haría, with 0.25% and 0.85% increases, respectively, and visits to these attractions were less likely per each extra income range.

Throughout this paper, the analysis of Table 2 allows for the effect of a variable on different attractions to be studied and for the effects of two or more variables on a specific attraction to be compared. Moreover, checking the results displayed in Table 2, it is also possible to analyze the effects of all the variables on just one attraction to characterize the profile of the tourists it receives. For instance, in the case of Jameos del Agua, older and richer tourists were less prone to visit the attraction, as already mentioned. It can also be noted that tourists' country of residence was important. Tourists living in Spain had greater odds of visiting this attraction, while tourists coming from Ireland, Norway, the United Kingdom, and Sweden were less likely to do so. Moreover, visits to all attractions included in the survey, except visits to San José Castle, had a positive effect on visits to Jameos del Agua. The motivations tourists had for travelling (holidays, business, etc.) did not seem to be significant to explain why Jameos del Agua was visited. However, the reasons why tourists had chosen Lanzarote as destination had a direct effect on the probabilities of visiting this attraction. For example, tourists who choose Lanzarote for its landscapes, relaxation, its culture, the possibility of practicing nautical activities or visiting new places were more likely to visit Jameos del Agua. However, choosing Lanzarote for its safety against natural disasters or because it offers the chance of doing sports not included in the survey (only golf and nautical activities were included) made it less likely to visit this attraction. Moreover, tourists who had previously visited the island were 1.11% less likely to

visit Jameos del Agua. Finally, each extra euro spent at the destination increased the probabilities of visiting this attraction, even though the effect was relatively small, under 0.01%. Indeed, spending on excursions and on cultural products increased the odds of going to Jameos del Agua by 0.11% and 0.13%, respectively, for each euro spent. However, each extra euro spent on paying for hotels, buses, taxis, sports activities, supermarkets, discos, souvenirs, restaurants, pharmacies, or other items not specified in the survey reduced the probabilities of visiting Jameos del Agua.

The analysis of Table 2 also allowed for the comparison of all the variables involved in visiting two or more attractions or even analyzing the effect of multiple variables on all the attractions studied. The case of country of residence is a good example to apply this, because very different behaviors can be seen in tourists' visiting patterns depending on the country they came from. Spanish tourists who travelled to Lanzarote were the only segment that had more probabilities of visiting Jameos del Agua. They were also more likely to visit Los Verdes Cave, Monument to the Peasant, and La Graciosa than other tourists, while they had fewer probabilities of visiting San José Castle and Cactus Garden.

McKercher and Mak [44] and Bi and Lehto [73], among others, already pointed to differences in tourists' behavior depending on their provenance. In fact, their findings indicated that domestic tourists have different mobility patterns to international visitors. However, behavioral differences were also found among international tourists in this research. Tourists from Germany and Austria showed similar results. They were more likely to visit the Monument to the Peasant and the César Manrique Foundation than other segments, but they had fewer probabilities of going to Timanfaya and San José Castle. However, there were some differences because German tourists were more likely to visit Los Verdes Cave and the markets of Teguise and Haría, while coming from Austria was not a significant variable to explain this. Conversely, Austrian tourists were more likely to go to Gazer del Río, but coming from Germany was not a significant variable to explain this behavior. The most similar behavior can be seen when analyzing the likelihood to visit attractions of tourists coming from Ireland and the United Kingdom, both being less likely than others to visit Jameos del Agua, Timanfaya, Gazer del Río, Cactus Garden, Arrecife, and La Graciosa. By analyzing the effect that country of residence has on visits to attractions, it was possible to see that this variable was more relevant for some attractions than for others. For example, in the case of the Cactus Garden, ten countries of residence could explain why this attraction is visited, but just two of them were significant to explain the visits made to the markets of Teguise and Haría. Moreover, for some attractions, the effect of the country of residence was a variable that just explained why the attraction was visited by certain segments, such as the Monument to the Peasant. In other cases, country of residence effects on probabilities were only negative for some attractions, such as Timanfaya. This means that tourists' residence can explain why places were not visited, but not the reasons why attractions were being visited. It is, therefore, necessary to consider other variables.

From the results obtained, it is possible to identify the most relevant variables that explain why the complete set of attractions was visited. While some variables, such as gender or motivation for travel, affected just a few attractions, others modified the probabilities of visiting several attractions. In particular, tourists' age, visits to other attractions, some countries of residence, certain motives to choose Lanzarote as a destination, purchasing all-inclusive packages, and specific expenditure items had an influence on the probabilities of visiting most of the attractions. Moreover, some variables had a higher effect on changing the probabilities of visiting certain attractions, while other variables had a much lower effect. For example, the variable regarding tourists who were motivated to travel to attend congresses and meetings shows that these tourists were 14.46% more likely to visit Gazer de Río. However, the variable regarding expenditure at destination shows that tourists were less than 0.01% more likely to visit several attractions per each extra euro spent. This indicates that given variables, despite being identified as significant by the analysis, are not very relevant to explain tourists' behavior. Some of the results obtained have similarities with previous studies, such as the ones carried out by Molinillo and Japutra [43] and Masiero and Zoltan [61], which identified being attracted by a destination's culture as a key variable to explain tourists' behavior. However, other results present

differences with previous literature, such as the study of Zakrisson and Zillinger [74], which indicated that travelling to visit friends and family is key to explaining why tourists do not visit attractions, while this variable was just significant in the current study to explain why just two out of 11 attractions were less likely to be visited.

In addition, the analysis also shows that the choice of certain attractions can be explained with a fewer number of variables, while others depend on a larger number of them. For example, visits made to San José Castle were explained by just 19 of the 86 possible explanatory variables. However, the visits made to La Graciosa were explained by 49 of the 86 variables that were found to be significant for the analysis. In spite of this, other variables that were not included in this study could increase the number of variables explaining why attractions were visited. Liu et al. [15] pointed out that attractions included in collaboration networks are more visited than attractions that are not included. Other elements, such as the spatial distribution of resources, attractions, and tourist establishments were also proven to have an influence on tourists' mobility patterns and visits. With this analysis, it was possible to analyze the attractions having an influence on the final assessment given by tourists of the trip. Timanfaya National Park was the most visited attraction analyzed and the models show that tourists who visited this attraction were more likely to give better assessments of their trips. It indicates that Timanfaya is a key tourism attraction for destination managers that must be carefully managed and preserved in order to continue attracting tourists to the destination [75]. It was also possible to confirm that La Graciosa and San José Castle, being the least visited attractions analyzed, are also key elements which tourism managers should focus on to increase tourists' satisfaction. In the case of San José Castle, visiting this attraction decreased the assessment given by tourists to their trips. This could explain why this attraction was just visited by 5% of tourists. Destination managers should improve the services offered at this attraction in order to increase the satisfaction of tourists visiting it and to increase the number of tourists willing to do so. In the case of La Graciosa, despite being one of the least visited attractions, tourists who visit it were more likely to give better assessments of their trips to Lanzarote. Nevertheless, La Graciosa is suffering from overtourism during the summer season, particularly because of the visits made by residents in the Canary Islands and the very limited capacity of absorption of its infrastructures given that the island is only 29 km$^2$ in size and has around 800 inhabitants. Recent research highlights the abundance of microplastics at an isolated beach in La Graciosa, an issue to be considered by tourism managers, even if this study does not attribute it to the tourist pressure [76].

The results of the analysis provide a segmentation of tourists based on their characteristics and their use of attractions and, at the same time, a segmentation of attractions based on the characteristics of tourists that visit them. This segmentation of the attraction market is crucial for designing policies to prevent overtourism. In fact, spreading tourism flows in time and space constitutes the first two strategies mentioned by the reports of UNWTO [24,25]. As suggested by Neuts and Nijkamp [26] and Oklevik et al. [22], the basis for this policy is the segmentation of tourists by their attributes and behavior.

## 5. Conclusions

In this paper, 37,055 tourist survey responses were analyzed to determine if the 96 variables selected from the survey could explain the visits made to 11 different tourist attractions on the island of Lanzarote. The use of discrete choice models to analyze the effects of all the variables on the attractions allow researchers to determine which variables are relevant to tourists' visits to each attraction. They also indicate how each variable modifies the probabilities of visiting the attractions. The results obtained through these models highlight that nine of the 96 variables were not significant to explain tourists' visits to the attractions studied. There were 13 variables that were significant to explain why just one of the attractions was visited. The analysis shows that several motives to choose the island as a destination (weather, environment, health tourism, theme parks, golf, easy transport, natural safety, and other motives) were only able to explain why one of the attractions

studied was visited. Another 33 variables were significant to explain why at least two attractions were visited, though they had different effects on the probabilities of visiting those attractions. One of these variables was tourists' age, which showed that older tourists had greater probabilities than younger ones of visiting cultural attractions or those attractions linked with Canary traditions. By contrast, each extra year of a tourist's age decreased their probabilities of visiting the most visited attractions of the destination. The other 40 variables were able to explain why at least two attractions were visited, and these variables had the same kind of effect on all these attractions. For example, Nordic and Anglo-Saxon tourists were usually less likely to visit the attractions analyzed, while variables regarding visits to other attractions commonly explained why a different attraction was visited. The existence of this high number of shared explanatory variables seems to highlight that there exists a general profile of a visitor who is more likely to visit all the attractions studied. Despite this, the 13 variables that were relevant to explain the visit to just one of the attractions and the 33 variables that showed to have different effects on the 11 attractions also point to the existence of certain features that make visitors likely to choose the attractions they prefer to visit.

Through the use of discrete choice models, it was also possible to establish which variables had a higher or lower effect on changing the probabilities of visiting attractions. This study also identified variables that had a similar effect on a set of attractions or even attractions that had more probabilities of being visited by tourists with certain characteristics. By doing so, this study allows segments to be identified in tourism demand and clustering the supply of attractions on the island of Lanzarote. This analysis also makes it possible to check the relationship between visiting an attraction and the increase in the amount of money spent during the trip or the assessment given by tourists of the trip. Moreover, it helped to identify the products which tourists who were interested in visiting an attraction were more likely to spend a larger amount of money on. The study of the relationship between visiting the main tourism attractions of the island and the assessment given to the trip by tourists can also help identify challenges faced by the destination [75]. Despite the results obtained in this study, more research on this topic is needed, as some results may differ from findings in previous studies. For example, the study of Zakrisson and Zillinger [74] pointed out that knowing if tourists are travelling to visit relatives can explain why tourists do not tend to visit attractions, while in this analysis, this variable showed to be significant for just two out of 11 attractions studied. Other studies have only pointed to behavioral differences between domestic and international tourists but, in this study, it was possible to identify differences depending on the country of residence [44,73].

Despite these implications and the large number of variables used, other variables not considered could have been analyzed to explain the behavior shown by tourists during their trips to Lanzarote. Using the hotel location, attraction prices, or the attractions included in packages sold by travel agencies and tour operators are examples of variables that could also have high importance in explaining why these 11 attractions were visited. Moreover, limiting the research to the 11 most visited attractions of the island omits analyzing why secondary attractions are visited or even knowing the characteristics of tourists who visit other frequented places, such as beaches. In addition, tourism attractions are not the only element of a destination that needs to be analyzed. The likelihood of tourists to choose accommodation, restaurants, or even their daily itineraries are factors that should be considered in future research. They can explain why certain attractions are visited and this can help destination management organizations in their decision-making processes. Additionally, it should be noted that only survey data was used during this research. Currently, data collection tools such as tracking devices or social media posts are being employed to gather information on tourists. The use of these tools could help analyze a larger number of attractions and explanatory variables that may provide more detailed information for improving decision-making by public and private tourism managers.

This research has implications for public and private tourism managers particularly in the context of preventing the effects of overtourism [77]. For destination management organizations, this research can be used to identify the characteristics of those tourists who are more prone to visit attractions and the effects that visiting attractions have on expenditure at the destination, on the assessment of the trip

given by tourists [78], and on the characteristics of tourists that can be encouraged to change their behavior with the purpose of spreading tourist flows throughout the island to prevent congestion in certain periods and places. These data can be used to identify a destination's weaknesses and strengths, which can then be used during decision-making processes to invest in certain infrastructures or to offer a wider set of services at certain locations. For tourism attraction managers, this analysis helps identify the characteristics of their main target segments, even when these segments do not represent the largest number of visitors received by the attraction. This allows marketing and promotional campaigns to focus on potential tourists with these characteristics or to change the management of the attraction's services to offer new products oriented to different segments. This kind of study is also useful to manage congestion within destinations or attractions, as they provide information on which tourists are more likely to visit certain attractions [35,79]. For destination managers that may not be interested in attracting a huge number of tourists to specific places to avoid congestion, particularly in sensitive environments like national parks, this analysis may help identify if the current profile of visitors received fits with the sustainable development policy of the destination. Thus, it may help to know if a destination is receiving tourists interested in its natural and cultural preservation or who show a higher tendency to spend larger amounts of money during their trips. This may allow for an optimization of tourism flows in the sense of trying to attract tourist that not only spend more, but also have a lower impact on the environment [22]. For other attractions or companies not included in this study, the results can point to the places visited by their main target segments. It can also help identify the characteristics of those tourists more likely to visit nearby attractions.

The results presented for each specific attraction are raw material for policy design to manage congestion and prevent overtourism. The recommendations of UNWTO [24] are to manage the negative impacts of tourist concentration, the relevance of diversifying demand in time and space and, at the same time, attracting the adequate tourist segments. The island of Lanzarote has been an international reference for tourism management. To take this management to the next level, a thorough analysis of tourists' characteristics who visit attractions based on survey information can be used as a basis for improvement in the design of a sustainability policy for tourism on this island for the coming years.

**Author Contributions:** Both authors have contributed to developing the main ideas of the paper, as well as the quantitative analysys, previous literature and writing. Hugo Padrón-Ávila has focused mainly on the determinants and the empirical aspects of tourists' visits to attractions, while Raúl Hernández-Martín has particularly developed the policy implications of the analysis in the context of sustainability and overtourism.

**Funding:** This research was supported and funded by the Government of the Canary Islands and the European Social Fund. The researchers thank the Canary Islands Institute of Statistics (ISTAC) and Lanzarote Data Center for providing the data needed to carry out this research.

**Conflicts of Interest:** The authors declare no conflict of interest. The sponsors had no role in the design, execution, interpretation, or writing of the study.

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
