# Peer review of "Preventing Overtourism by Identifying the Determinants of Tourists’ Choice of Attractions"

_sustainability, doi:10.3390/su11195177_

Round 1

Reviewer 1 Report

The manuscript addresses an interesting topic. You need to make revisions to your paper before I can recommend the paper. My comments are listed below for your reference: 

Introduction:

Redevelop your introduction to address the importance of your research in relation to sustainability of management of attractions. What you said in the 2nd paragraph is better in your methodology instead of your introduction. You need to explain your research aim and objectives. You need to introduce the structure of your paper. You explain the importance of your research in the 3rd paragraph and such paragraph could be more concise. Practical importance could be highlighted when you discuss practical implications of your research.

Background

Background section is better called as ‘Literature review’ based on what your content is. When you discuss previous literature, different tenses (past, present and present perfect) are used. You need to use ONE consistent tense. Multiple citations should be provided for lines 82 to 85. A table could be provided to summarise factors influencing tourists’ choice of attractions based on the review of relevant literature sources after Line 106.

Methods:

Great to see a huge sample size is used for this research and longitudinal data as well. More could be said pilot study and sampling procedures.

Results:

Your data are thoroughly analysed and explained. However, there is no discussion of your results in relation to previous literature sources. Last three paragraphs in this section could be adapted to your discussion. However, you need to add more sentences to compare them with previous literature sources and provide some explanation to the dissimilarity.

Conclusion

The first paragraph could be more concise to only include key findings. When you discuss practical implications, it would be better if you can make further linkage with management of attractions in a sustainable manner. The third paragraph is fine.

Author Response

Introduction:

Redevelop your introduction to address the importance of your research in relation to sustainability of management of attractions.

Both authors thank the reviewer for his comments, which have allowed to improve the quality of the study and the writing of the paper.

As the reviewer points, intention of the authors is publishing in a special issue edition about overtourism and revitalization, so the possible applications of the study to help DMO manage these issues have been added to the introduction:

Current media and academic debates have highlighted increasing concerns over the negative impact of the concentration of tourism flows in certain areas, particularly in cities, in a phenomenon that has been labeled as overtourism. Overtourism appears to be closely linked with the popularity of particular attractions and points of interest that have become ‘hotspots’ suffering from congestion [1]. However, effective attraction management can improve tourists’ satisfaction and destination sustainability, avoiding the negative impacts of the growth of tourist flows. The effects of overtourism can also be often suffered on small tourism islands, where land availability is constrained [1].

The island of Lanzarote has an area of 846 km² and is the easternmost island of the Canary Island autonomous region in Spain. The mild climate of these islands allows tourist arrivals throughout the whole year without a marked seasonality pattern. Lanzarote is the third main tourism island of the Canary Island, which is the region with the highest figures of overnight stays in tourism accommodation in Europe [2]. Using data published by the Canary Islands Institute of Statistics for 2018, Lanzarote has a population of just 149 thousand inhabitants and yet received 2.9 million tourists from outside the archipelago with an average length of stay of 7.8 days [3]. Lanzarote was declared UNESCO Biosphere Reserve in 1993. Two years after this, the island hosted the World Conference on Sustainable Tourism [4], as part of its commitment to responsible tourism. However, the island has experienced tensions since the early economic growth of the 1960s between tourism development and the island’s natural and cultural preservation [5]. In fact, the debate on the limits of growth and sustainability begun in the Canary Islands four decades ago, and especially in Lanzarote, this debate has been part of its strategy for tourism development for a long time. These concerns resulted in the implementation in 2001 of a moratorium on new hotels in the Canary Islands [6]. Despite these restrictions on the growth of the accommodation supply, tourism arrivals continued to increase in Lanzarote along with population figures (55% increase in inhabitants from 2000 to 2018) [3]. Therefore, public concerns with congestion and the negative impacts of rapid growth have been recurrent in the Canary Islands long before the recent renewed interest in tourism impacts under the concept of overtourism [7]. The interest of Lanzarote in sustainable tourism has been recognized widely and recently highlighted by research on sustainable tourism mobility on the island [8]. Furthermore, Eckert and Pechlaner have recognized the conflict between sustainability and tourism growth in Lanzarote and have made a proposal for product and target group market diversification [9]. These kinds of policies are closely related to the segmentation of tourists to improve destination management and, consequently, to the aims of this research.

The tourism model of Lanzarote is somewhat particular within the Canary Islands. Besides sun and sand facilities, the destination management organization is involved in providing a set of outstanding natural and cultural attractions to be visited by tourists. These attractions are mainly managed by a public-owned firm (Centros de Arte, Cultura y Turismo de Lanzarote, Arts, Culture and Tourism Centers of Lanzarote), which depends on the Lanzarote Island Council. This set of attractions including a volcanic national park, volcanic caves, cultural heritage attractions, etc. received 3 million visits in 2018, with a moderate 9.2% increase in visits to the main attractions between 2000 and 2018. Arts, Culture and Tourism Centers of Lanzarote obtained 30 million € turnover in 2018 of which 67.3% came from selling tickets, 20.7% from restaurant sales and 11.1% from sales made in the shops of the attractions [10].

As mentioned, concerns about congestion arose in Lanzarote at least four decades ago, long before the recent renewed academic interest in the negative effects of rapid tourism growth, fostered by low cost airlines, the sharing economy, new booking platforms, social networks, etc. put the topic in the media and academic spotlight [11]. However, the island of Lanzarote can be an example of successful public management of a destination and its attractions. In fact, a survey on ‘Island Affairs’ made in December 2017 for the UNESCO Biosphere Reserve shows a remarkable improvement from 2006 to 2017 in inhabitants’ perceptions about the environmental situation of the island [12].

What you said in the 2nd paragraph is better in your methodology instead of your introduction. You need to explain your research aim and objectives. You need to introduce the structure of your paper.

The second paragraph has been summarized in order not to include so many methodological aspects. Instead of this, the structure of the paper has been written to help readers understand its parts and how it has been divided. In addition, the aim of the paper has been clearly stated. The paragraph has been modified as follows:

In this paper, the aim is to determine which tourist characteristics make them more likely to visit certain attractions instead of others. This information is crucial for the management of attractions to prevent overtourism. To do so, several binomial discrete choice models have been used. These provide an understanding of how a set of 96 different variables impact the probabilities of visiting 11 different attractions.

First, a literature review is carried out to discover which variables have been pointed out by other researchers as determinants to explain tourists’ visits to attractions , methods used to analyze these data and the relevance of the concept of overtourism in the context of a mass tourism destination based on attractions. After this, the data and the method chosen to carry out the study are explained. Afterwards, the results of the research are presented in a table that contains data regarding the extent that the variables used change the probabilities of visiting each of the 11 attractions studied. The data showed in the table are discussed to establish which variables explain why each of the attractions is visited or not. Finally, the conclusions of the research are presented and the results obtained are compared with previous research. Knowing which variables explain visits to attractions is useful for destination managers aiming to prevent overtourism and to design revitalization policies.

You explain the importance of your research in the 3rd paragraph and such paragraph could be more concise. Practical importance could be highlighted when you discuss practical implications of your research.

The third paragraph of the previous manuscript has been deleted to substitute it for a first introduction paragraph indicating the relationship between the paper and the management of overtourism and revitalization. So, this paragraph has been added:

Current media and academic debates have highlighted increasing concerns over the negative impact of the concentration of tourism flows in certain areas, particularly in cities, in a phenomenon that has been labeled as overtourism. Overtourism appears to be closely linked with the popularity of particular attractions and points of interest that have become ‘hotspots’ suffering from congestion [1]. However, effective attraction management can improve tourists’ satisfaction and destination sustainability, avoiding the negative impacts of the growth of tourist flows. The effects of overtourism can also be often suffered on small tourism islands, where land availability is constrained [1].

Background:

Background section is better called as ‘Literature review’ based on what your content is. When you discuss previous literature, different tenses (past, present and present perfect) are used. You need to use ONE consistent tense.

Thanks for these comments. Effectively, the use of different tenses was confusing, so it has been properly changed for past tenses in all cases it was possible. Moreover, this section has been called “Literature review”.

Multiple citations should be provided for lines 82 to 85.

Citations have been added to the sentence indicated, so it has been modified as follows:

Certain studies have highlighted the relevance of the socio-economic characteristics of visitors and attractions when it comes to explaining the type of places visited by tourists [34–36].

A table could be provided to summarise factors influencing tourists’ choice of attractions based on the review of relevant literature sources after Line 106.

The table suggested by the reviewer has been added to the literature review as requested, indicating the variable influenced and the bibliographic references indicating this influence. So, this table has been added:

Table 1. List of variables explaining tourists’ choices.

Variable

Reference

Socio-economic characteristics

[34–36]

Age

[38]

Gender

[38]

Country of residence

[41–44]

Sexual orientation

[39,40]

Motivations

[27,28]

Relaxation

[28]

Culture

[29,30]

MICE tourism

[32]

Health

[33]

Characteristics of attractions

[45,46]

Price

[37]

Second homes

[31]

Accommodation location

[16]

Methods:

Great to see a huge sample size is used for this research and longitudinal data as well. More could be said pilot study and sampling procedures.

Both authors thank the reviewer for appreciating the database used and the analysis made. A pilot study was indeed made before carrying out the current study and data were filtered in order to omit missing data in the variables regarding visits to attractions. So, these paragraphs were modified in the “Method” section of the paper:

First, a pilot study using contingency tables was applied to know which variables could explain whether the attractions analyzed were visited or not. Afterwards, 96 variables were chosen to carry out the analysis based on the results obtained and the literature review presented in the previous section. The choice of the variables has been based on the research by Padrón-Ávila & Hernández-Martín [68]. The variables chosen were: tourists’ gender, age, annual incomes, country of residence, main reasons for travelling, motives for choosing Lanzarote as destination, overnight stays, travel group, purchase of low-cost flights, purchase of all-inclusive packages, repetition at the destination, total expenditure at origin, total expenditure at the destination, practice of diving, and overall assessment of the trip. Moreover, the 11 variables regarding visits to tourist attractions in Lanzarote were used to determine if the visit to a given attraction is linked with a visit to others. Most variables used in the analysis are binary variables (take the values 0 or 1). In the case of tourists’ country of residence, 20 binary variables attached to different countries have been used. In the case of main reasons for travelling, 6 binary variables have been used to indicate if tourists are travelling for holidays, business, attending conferences, health, visiting friends and relatives, or other motives. In the case of motives to choose Lanzarote as destination, 23 different motives have been considered and tourists could indicate up to three in the survey. In the case of travel group, 6 different groups have been considered, as tourists could indicate if they were travelling with their partner, with children, with other relatives, with friends, with or with other companions. Variables regarding tourists’ age, overnights stays and expenditure are continuous. In the case of expenditure at the destination, this variable was divided into 19 items regarding different services and products typically bought by tourists. Moreover, tourists were divided into seven discrete ranges according to their annual incomes, so a lower value in this variable implies a lower income perceived by the respondent. In addition, a Likert scale of five items was used to measure the variable regarding the assessment of the trip made.

The complete database given by ISTAC contains 257,687 survey responses gathered from tourists visiting any of the Canary Islands. As this research focuses on 11 attractions located in the island of Lanzarote, a total of 43,301 responses of tourists that were surveyed from January 2010 to December 2016 at the airport of Lanzarote were considered. However, 14.42% of tourists surveyed in Lanzarote did not respond to the question regarding the attractions visited or not visited in the island. Therefore, these responses were not considered when analyzing an attraction for which the corresponding data had not been provided. Regarding the attractions analyzed, four of them were visited by more than a quarter of tourists surveyed. There were 48% of tourists who visited the Timanfaya National Park, 37% visited Jameos del agua, 31% went to Gazer del Río and 29% went to Los Verdes Cave. Five attractions were visited by 10% to 25% of tourists: these attractions were the Cactus Garden, Arrecife Town, the traditional markets (these ones were visited by 21% of tourists), the Monument to the Peasant (14%) and the César Manrique Foundation (13%). San José Castle and La Graciosa were visited by just 5% of tourists surveyed.

The Survey on Tourism Expenditure of the Canary Islands includes 43,301 responses to the survey in the island of Lanzarote from January 2010 to December 2016. The survey is conducted in the departure hall of the airport following international recommendations for tourism expenditure statistics [69] and it uses non-probability stratified sampling in three stages. This method does not allow obtaining the sample errors. In the first stage of the sampling process, the week with more air traffic is chosen in each month of the year. In a second stage, stratified sampling allows to choose the countries and airports of origin of tourists and the timetables. In the third stage, tourists from mainland Spain and foreign countries are selected [70].

Literature in sampling states that biases and sample variance are the two possible errors when using non-probability sampling. To prevent these errors from affecting the results, the methodology applied in the Survey on Tourism Expenditure of the Canary Islands includes elements of control and correction. To avoid biases, a post- stratification weighting is conducted with the help of auxiliary information to correct biases in age, gender, country of residence and type of accommodation; thus reliability of estimates and inferences is increased [71]. To control for sampling variance, a minimum of 20 observations is required for each variable. This threshold settles a maximum admissible error under assumptions of probabilistic sampling and maximum population variance. In this study, with a sample of 43,301 this threshold is exceeded by far in the responses.

Results:

Your data are thoroughly analysed and explained. However, there is no discussion of your results in relation to previous literature sources. Last three paragraphs in this section could be adapted to your discussion. However, you need to add more sentences to compare them with previous literature sources and provide some explanation to the dissimilarity.

Certainly, the discussion of the results was missing in this paper and it was necessary to add it to compare the results obtained with previous studies to establish similarities and differences. Thus, the following sentences have been added in “Results” and “Conclusions” sections to include some discussion on this topic:

In fact, Lew & McKercher [18] have already pointed out that tourists purchasing all-inclusive packages are less likely to visit attractions.

In fact, previous studies have already indicated that the kind of companions during a trip is key to determine which attractions will be visited [72].

McKercher & Mak [44] and Bi & Lehto [73], among others, have already pointed to differences in tourists’ behavior depending on their provenance. In fact, their findings indicated that domestic tourists have different mobility patterns to international visitors. However, behavior differences have also been found among international tourists in this research.

Some of the results obtained have similarities with previous studies, such as the ones carried out by Molinillo & Japutra [43] and Masiero & Zoltan [61], which identified being attracted by a destination’s culture as a key variable to explain tourists’ behavior. However, other results present differences with previous literature, such as the study of Zakrisson & Zillinger [74], which indicated that travelling to visit friends and family is key to explaining why tourists do not visit attractions, while this variable was just significant in the current study to explain why just 2 out of 11 attractions were less likely to be visited.

In spite of this, other variables that have not been included in this study could increase the number of variables explaining why attractions are visited. Liu et al. [15] pointed that attractions included in collaboration networks are more visited than not included attractions. Other elements, such as the spatial distribution of resources, attractions and tourist establishments have also been proven to have an influence on tourists’ mobility patterns and visits.

Despite the results obtained in this study, more research on this topic is needed as some results may differ from findings in previous studies. For example, the study of Zakrisson & Zillinger [74] pointed out that knowing if tourists are travelling to visit relatives can explain why tourists do not tend to visit attractions, while in this analysis, this variable showed to be significant for just 2 out of 11 attractions studied. Other studies have only pointed to behavioral differences between domestic and international tourists but, in this study, it has been possible to identify differences depending on the country of residence [44,73].

Conclusion:

The first paragraph could be more concise to only include key findings. When you discuss practical implications, it would be better if you can make further linkage with management of attractions in a sustainable manner. The third paragraph is fine.

First paragraph has been modified to also included key findings of the results, so it can be used as a summary of the research carried out. The paragraph has ended as follows:

In this paper, 37,055 tourist survey responses have been analyzed to determine if the 96 variables selected from the survey could explain the visits made to 11 different tourist attractions on the island of Lanzarote. The use of discrete choice models to analyze the effects of all the variables on the attractions allow researchers to determine which variables are relevant to tourists’ visits to each attraction. They also indicate how each variable modifies the probabilities of visiting the attractions. The results obtained through these models highlight that 9 of the 96 variables were not significant to explain tourists’ visits to the attractions studied. There were 13 variables that were significant to explain why just one of the attractions was visited. The analysis shows that several motives to choose the island as destination (weather, environment, health tourism, theme parks, golf, easy transport, natural safety and other motives) were only able to explain why one of the attractions studied was visited. Another 33 variables were significant to explain why at least two attractions were visited, though they had different effects on the probabilities of visiting those attractions. One of these variables was tourists’ age, which showed that older tourists have greater probabilities than younger ones of visiting cultural attractions or those attractions linked with Canary traditions. By contrast, each extra year-old tourists are decreases their probabilities of visiting the most visited attractions of the destination. The other 40 variables were able to explain why at least two attractions were visited, and these variables had the same kind of effect on all these attractions. For example, Nordic and Anglo-Saxon tourists are usually less likely to visit the attractions analyzed, while variables regarding visits to other attractions commonly explains why a different attraction is visited. The existence of this high number of shared explanatory variables seems to highlight that there exists a general profile of visitor who is more likely to visit all the attractions studied. Despite this, the 13 variables that were relevant to explain the visit to just one of the attractions and the 33 variables that showed to have different effects on the 11 attractions also point to the existence of certain features that make visitors likely to choose the attractions they prefer to visit.

Certainly, a deeper analysis regarding implications of the study for managing sustainable attractions is needed as this paper would be published in a special issue about managing overtourism and revitalization. Thus, these implications have been added to the conclusions:

This research has implications for public and private tourism managers particularly in the context of preventing the effects of overtourism [77]. For destination management organizations, this research can be used to identify the characteristics of those tourists who are more prone to visit attractions and the effects that visiting attractions have on expenditure at the destination, on the assessment of the trip given by tourists [78] and on the characteristics of tourists that can be encouraged to change their behavior with the purpose of spreading tourist flows throughout the island to prevent congestion in certain periods and places. These data can be used to identify a destination’s weaknesses and strengths, which can then be used during decision-making processes to invest in certain infrastructures or to offer a wider set of services at certain locations. For tourism attraction managers, this analysis helps identify the characteristics of their main target segments, even when these segments do not represent the largest number of visitors received by the attraction. This allows marketing and promotional campaigns to focus on potential tourists with these characteristics or to change the management of the attraction’s services to offer new products oriented to different segments. This kind of study is also useful to manage congestion within destinations or attractions, as they provide information on which tourists are more likely to visit certain attractions [35,79]. For destination managers that may not be interested in attracting a huge number of tourists to specific places to avoid congestion, particularly in sensitive environments like national parks, this analysis may help identify if the current profile of visitors received fits with the sustainable development policy of the destination. Thus, it may help to know if a destination is receiving tourists interested in its natural and cultural preservation or who show a higher tendency to spend larger amounts of money during their trips. This may allow for an optimization of tourism flows in the sense of trying to attract tourist that not only spend more, but also have a lower impact on the environment [22]. For other attractions or companies not included in this study, the results can point to the places visited by their main target segments. It can also help identify the characteristics of those tourists more likely to visit nearby attractions.

The results presented for each specific attraction is raw material for policy design to manage congestion and prevent overtourism. The recommendations of UNWTO [24] are to manage the negative impacts of tourist concentration stress, the relevance of diversifying demand in time and space and, at the same time, attracting the adequate tourist segments. The island of Lanzarote has been an international reference for tourism management. To take this management to the next level, a thorough analysis of tourists’ characteristics who visit attractions based on survey information can be used as a basis for improvement in the design of a sustainability policy for tourism on this island for the coming years.

Reviewer 2 Report

Dear Author/s:

Being an interesting manuscript “Tourists’ Characteristics and Their Choice of Attraction “., based on a solid investigation it has the necessary merits to be published in Sustainability. However, there are some issues that should be addressed prior to publication.

1. Introduction (line 30)

Authors presents the importance  of the study about t“Tourists’ Characteristics , as well as, the main objective pursued. However, it would be convenient to include the secondary objectives that are pursued with the realization of research.

2. Literature Review and Hypothesis Development

The review is adequate.

3. Methods  (line 140)

Correct method (Mehtod, line 140)

In the Materials and Methods section (line 140) please add more information about the quantitative survey (Questionnaire and scales)

The authors should expand the Analysis of average the partial effect models used. Measurement model (line 222, table 1). A conceptual explanation should be provided for variables presented

4. Conclusions (line 361)

Authors should have more discussion in the conclusions.

Author Response

Introduction (line 30)

Authors presents the importance of the study about Tourists’ Characteristics, as well as, the main objective pursued. However, it would be convenient to include the secondary objectives that are pursued with the realization of research.

Both authors thank the reviewer for his comments, which have allowed to improve the quality of the study and the writing of the paper.

As the reviewer points, no secondary objectives had been added to the study. As the intention of the authors is publishing in a special issue edition about overtourism and revitalization, the possible applications of the study to help DMO manage these issues have been added as secondary objectives:

Current media and academic debates have highlighted increasing concerns over the negative impact of the concentration of tourism flows in certain areas, particularly in cities, in a phenomenon that has been labeled as overtourism. Overtourism appears to be closely linked with the popularity of particular attractions and points of interest that have become ‘hotspots’ suffering from congestion [1]. However, effective attraction management can improve tourists’ satisfaction and destination sustainability, avoiding the negative impacts of the growth of tourist flows. The effects of overtourism can also be often suffered on small tourism islands, where land availability is constrained [1].

The island of Lanzarote has an area of 846 km² and is the easternmost island of the Canary Island autonomous region in Spain. The mild climate of these islands allows tourist arrivals throughout the whole year without a marked seasonality pattern. Lanzarote is the third main tourism island of the Canary Island, which is the region with the highest figures of overnight stays in tourism accommodation in Europe [2]. Using data published by the Canary Islands Institute of Statistics for 2018, Lanzarote has a population of just 149 thousand inhabitants and yet received 2.9 million tourists from outside the archipelago with an average length of stay of 7.8 days [3]. Lanzarote was declared UNESCO Biosphere Reserve in 1993. Two years after this, the island hosted the World Conference on Sustainable Tourism [4], as part of its commitment to responsible tourism. However, the island has experienced tensions since the early economic growth of the 1960s between tourism development and the island’s natural and cultural preservation [5]. In fact, the debate on the limits of growth and sustainability begun in the Canary Islands four decades ago, and especially in Lanzarote, this debate has been part of its strategy for tourism development for a long time. These concerns resulted in the implementation in 2001 of a moratorium on new hotels in the Canary Islands [6]. Despite these restrictions on the growth of the accommodation supply, tourism arrivals continued to increase in Lanzarote along with population figures (55% increase in inhabitants from 2000 to 2018) [3]. Therefore, public concerns with congestion and the negative impacts of rapid growth have been recurrent in the Canary Islands long before the recent renewed interest in tourism impacts under the concept of overtourism [7]. The interest of Lanzarote in sustainable tourism has been recognized widely and recently highlighted by research on sustainable tourism mobility on the island [8]. Furthermore, Eckert and Pechlaner have recognized the conflict between sustainability and tourism growth in Lanzarote and have made a proposal for product and target group market diversification [9]. These kinds of policies are closely related to the segmentation of tourists to improve destination management and, consequently, to the aims of this research.

The tourism model of Lanzarote is somewhat particular within the Canary Islands. Besides sun and sand facilities, the destination management organization is involved in providing a set of outstanding natural and cultural attractions to be visited by tourists. These attractions are mainly managed by a public-owned firm (Centros de Arte, Cultura y Turismo de Lanzarote, Arts, Culture and Tourism Centers of Lanzarote), which depends on the Lanzarote Island Council. This set of attractions including a volcanic national park, volcanic caves, cultural heritage attractions, etc. received 3 million visits in 2018, with a moderate 9.2% increase in visits to the main attractions between 2000 and 2018. Arts, Culture and Tourism Centers of Lanzarote obtained 30 million € turnover in 2018 of which 67.3% came from selling tickets, 20.7% from restaurant sales and 11.1% from sales made in the shops of the attractions [10].

As mentioned, concerns about congestion arose in Lanzarote at least four decades ago, long before the recent renewed academic interest in the negative effects of rapid tourism growth, fostered by low cost airlines, the sharing economy, new booking platforms, social networks, etc. put the topic in the media and academic spotlight [11]. However, the island of Lanzarote can be an example of successful public management of a destination and its attractions. In fact, a survey on ‘Island Affairs’ made in December 2017 for the UNESCO Biosphere Reserve shows a remarkable improvement from 2006 to 2017 in inhabitants’ perceptions about the environmental situation of the island [12].

In this paper, the aim is to determine which tourist characteristics make them more likely to visit certain attractions instead of others. This information is crucial for the management of attractions to prevent overtourism. To do so, several binomial discrete choice models have been used. These provide an understanding of how a set of 96 different variables impact the probabilities of visiting 11 different attractions.

Literature Review and Hypothesis Development

The review is adequate.

We thank the reviewer again for appreciating the work done.

Methods (line 140)

Correct method (Mehtod, line 140)

Thanks for pointing this mistake. The text was revised by a native English speaker, but these mistakes are usually disregarded. It has been corrected.

In the Materials and Methods section (line 140) please add more information about the quantitative survey (Questionnaire and scales)

As more information could be needed by future readers to understand the survey used to gather data and how these data were processed, more information regarding these topics has been added. Thus, these paragraphs have been added/modified:

First, a pilot study using contingency tables was applied to know which variables could explain whether the attractions analyzed were visited or not. Afterwards, 96 variables were chosen to carry out the analysis based on the results obtained and the literature review presented in the previous section. The choice of the variables has been based on the research by Padrón-Ávila & Hernández-Martín [68]. The variables chosen were: tourists’ gender, age, annual incomes, country of residence, main reasons for travelling, motives for choosing Lanzarote as destination, overnight stays, travel group, purchase of low-cost flights, purchase of all-inclusive packages, repetition at the destination, total expenditure at origin, total expenditure at the destination, practice of diving, and overall assessment of the trip. Moreover, the 11 variables regarding visits to tourist attractions in Lanzarote were used to determine if the visit to a given attraction is linked with a visit to others. Most variables used in the analysis are binary variables (take the values 0 or 1). In the case of tourists’ country of residence, 20 binary variables attached to different countries have been used. In the case of main reasons for travelling, 6 binary variables have been used to indicate if tourists are travelling for holidays, business, attending conferences, health, visiting friends and relatives, or other motives. In the case of motives to choose Lanzarote as destination, 23 different motives have been considered and tourists could indicate up to three in the survey. In the case of travel group, 6 different groups have been considered, as tourists could indicate if they were travelling with their partner, with children, with other relatives, with friends, with or with other companions. Variables regarding tourists’ age, overnights stays and expenditure are continuous. In the case of expenditure at the destination, this variable was divided into 19 items regarding different services and products typically bought by tourists. Moreover, tourists were divided into seven discrete ranges according to their annual incomes, so a lower value in this variable implies a lower income perceived by the respondent. In addition, a Likert scale of five items was used to measure the variable regarding the assessment of the trip made.

The complete database given by ISTAC contains 257,687 survey responses gathered from tourists visiting any of the Canary Islands. As this research focuses on 11 attractions located in the island of Lanzarote, a total of 43,301 responses of tourists that were surveyed from January 2010 to December 2016 at the airport of Lanzarote were considered. However, 14.42% of tourists surveyed in Lanzarote did not respond to the question regarding the attractions visited or not visited in the island. Therefore, these responses were not considered when analyzing an attraction for which the corresponding data had not been provided. Regarding the attractions analyzed, four of them were visited by more than a quarter of tourists surveyed. There were 48% of tourists who visited the Timanfaya National Park, 37% visited Jameos del agua, 31% went to Gazer del Río and 29% went to Los Verdes Cave. Five attractions were visited by 10% to 25% of tourists: these attractions were the Cactus Garden, Arrecife Town, the traditional markets (these ones were visited by 21% of tourists), the Monument to the Peasant (14%) and the César Manrique Foundation (13%). San José Castle and La Graciosa were visited by just 5% of tourists surveyed.

The Survey on Tourism Expenditure of the Canary Islands includes 43,301 responses to the survey in the island of Lanzarote from January 2010 to December 2016. The survey is conducted in the departure hall of the airport following international recommendations for tourism expenditure statistics [69] and it uses non-probability stratified sampling in three stages. This method does not allow obtaining the sample errors. In the first stage of the sampling process, the week with more air traffic is chosen in each month of the year. In a second stage, stratified sampling allows to choose the countries and airports of origin of tourists and the timetables. In the third stage, tourists from mainland Spain and foreign countries are selected [70].

Literature in sampling states that biases and sample variance are the two possible errors when using non-probability sampling. To prevent these errors from affecting the results, the methodology applied in the Survey on Tourism Expenditure of the Canary Islands includes elements of control and correction. To avoid biases, a post- stratification weighting is conducted with the help of auxiliary information to correct biases in age, gender, country of residence and type of accommodation; thus reliability of estimates and inferences is increased [71]. To control for sampling variance, a minimum of 20 observations is required for each variable. This threshold settles a maximum admissible error under assumptions of probabilistic sampling and maximum population variance. In this study, with a sample of 43,301 this threshold is exceeded by far in the responses.

The authors should expand the Analysis of average the partial effect models used. Measurement model (line 222, table 1). A conceptual explanation should be provided for variables presented.

In the case of giving a better explanation of how average partial effects work, this paragraph has been modified to give a better explanation of their use and interpretation:

Once the discrete choice models were calculated and the significant variables to explain tourists’ behavior had been identified, average partial effects of these variables were calculated for each attraction. Calculating the average partial effects of these models, the results show the increase or decrease in the probability a tourist has of visiting an attraction when the tourist shows value 1 for one of the binary variables analyzed [59,64,65]. In addition, when analyzing other kinds of variables (continuous, Likert scale or divided by ranges), the results indicate how the average probabilities of visiting an attraction are modified by each unit added to the variables studied. For example, in the case of age, results of average partial effects show to what extent probabilities of visiting an attraction are modified by each extra year old a tourist is, taking 16 years old as starting point as this was the age of youngest tourists surveyed. If the coefficient resulting from average partial effects is positive, it means that each extra unit added to the variable increases the probabilities of visiting the attraction analyzed. While the probabilities will be reduced by each extra unit added to the variable if the result is negative. To carry out both analyses (discrete choice models and average partial effects), the software Stata version 12 was used. As the results of partial average effects are coefficients, these coefficients have been converted into percentages in this paper to facilitate the interpretation of the results.

As the reviewer has also pointed out that more information about the variables is needed, short notes have been added to indicate variables’ scales and to describe the variables included. This paragraph has been modified to add this information:

First, a pilot study using contingency tables was applied to know which variables could explain whether the attractions analyzed were visited or not. Afterwards, 96 variables were chosen to carry out the analysis based on the results obtained and the literature review presented in the previous section. The choice of the variables has been based on the research by Padrón-Ávila & Hernández-Martín [68]. The variables chosen were: tourists’ gender, age, annual incomes, country of residence, main reasons for travelling, motives for choosing Lanzarote as destination, overnight stays, travel group, purchase of low-cost flights, purchase of all-inclusive packages, repetition at the destination, total expenditure at origin, total expenditure at the destination, practice of diving, and overall assessment of the trip. Moreover, the 11 variables regarding visits to tourist attractions in Lanzarote were used to determine if the visit to a given attraction is linked with a visit to others. Most variables used in the analysis are binary variables (take the values 0 or 1). In the case of tourists’ country of residence, 20 binary variables attached to different countries have been used. In the case of main reasons for travelling, 6 binary variables have been used to indicate if tourists are travelling for holidays, business, attending conferences, health, visiting friends and relatives, or other motives. In the case of motives to choose Lanzarote as destination, 23 different motives have been considered and tourists could indicate up to three in the survey. In the case of travel group, 6 different groups have been considered, as tourists could indicate if they were travelling with their partner, with children, with other relatives, with friends, with or with other companions. Variables regarding tourists’ age, overnights stays and expenditure are continuous. In the case of expenditure at the destination, this variable was divided into 19 items regarding different services and products typically bought by tourists. Moreover, tourists were divided into seven discrete ranges according to their annual incomes, so a lower value in this variable implies a lower income perceived by the respondent. In addition, a Likert scale of five items was used to measure the variable regarding the assessment of the trip made.

Conclusions (line 361)

Authors should have more discussion in the conclusions.

Certainly, the discussion of the results was missing in this paper and it was necessary to add it to compare the results obtained with previous studies to establish similarities and differences. Thus, the following sentences have been added in “Results” and “Conclusions” sections to include some discussion on this topic:

In fact, Lew & McKercher [18] have already pointed out that tourists purchasing all-inclusive packages are less likely to visit attractions.

In fact, previous studies have already indicated that the kind of companions during a trip is key to determine which attractions will be visited [72].

McKercher & Mak [44] and Bi & Lehto [73], among others, have already pointed to differences in tourists’ behavior depending on their provenance. In fact, their findings indicated that domestic tourists have different mobility patterns to international visitors. However, behavior differences have also been found among international tourists in this research.

Some of the results obtained have similarities with previous studies, such as the ones carried out by Molinillo & Japutra [43] and Masiero & Zoltan [61], which identified being attracted by a destination’s culture as a key variable to explain tourists’ behavior. However, other results present differences with previous literature, such as the study of Zakrisson & Zillinger [74], which indicated that travelling to visit friends and family is key to explaining why tourists do not visit attractions, while this variable was just significant in the current study to explain why just 2 out of 11 attractions were less likely to be visited.

In spite of this, other variables that have not been included in this study could increase the number of variables explaining why attractions are visited. Liu et al. [15] pointed that attractions included in collaboration networks are more visited than not included attractions. Other elements, such as the spatial distribution of resources, attractions and tourist establishments have also been proven to have an influence on tourists’ mobility patterns and visits.

Despite the results obtained in this study, more research on this topic is needed as some results may differ from findings in previous studies. For example, the study of Zakrisson & Zillinger [74] pointed out that knowing if tourists are travelling to visit relatives can explain why tourists do not tend to visit attractions, while in this analysis, this variable showed to be significant for just 2 out of 11 attractions studied. Other studies have only pointed to behavioral differences between domestic and international tourists but, in this study, it has been possible to identify differences depending on the country of residence [44,73].